# LEGO-Learn: Label-Efficient Graph Open-Set Learning

**Haoyan Xu**  *haoyanxu@usc.edu*
*University of Southern California*

**Kay Liu**  *zliu234@uic.edu*
*University of Illinois Chicago*

**Zhengtao Yao**  *zyao9248@usc.edu*
*University of Southern California*

**Philip S. Yu**  *psyu@uic.edu*
*University of Illinois Chicago*

**Mengyuan Li**  *mengyuanli@usc.edu*
*University of Southern California*

**Kaize Ding**  *kaize.ding@northwestern.edu*
*Northwestern University*

**Yue Zhao**  *yzhao010@usc.edu*
*University of Southern California*

**Reviewed on OpenReview:** *https://openreview.net/forum?id=J6oxTJPOyN*

## Abstract

How can we train graph-based models to recognize unseen classes while keeping labeling costs low? Graph open-set learning (GOL) and out-of-distribution (OOD) detection aim to address this challenge by training models that can accurately classify known, in-distribution (ID) classes while identifying and handling previously unseen classes during inference. It is critical for high-stakes, real-world applications where models frequently encounter unexpected data, including finance, security, and healthcare. However, current GOL methods assume access to a large number of labeled ID samples, which is unrealistic for large-scale graphs due to high annotation costs.

In this paper, we propose LEGO-Learn (Label-Efficient Graph Open-set Learning), a novel framework that addresses open-set node classification on graphs within a given label budget by selecting the most informative ID nodes. LEGO-Learn employs a GNN-based filter to identify and exclude potential OOD nodes and then selects highly informative ID nodes for labeling using the K-Medoids algorithm. To prevent the filter from discarding valuable ID examples, we introduce a classifier that differentiates between the $C$ known ID classes and an additional class representing OOD nodes (hence, a $C + 1$ classifier). This classifier utilizes a weighted cross-entropy loss to balance the removal of OOD nodes while retaining informative ID nodes. Experimental results on four real-world datasets demonstrate that LEGO-Learn significantly outperforms leading methods, achieving up to a 6.62% improvement in ID classification accuracy and a 7.49% increase in AUROC for OOD detection.

## 1 Introduction

Graph-structured data has become increasingly important in various fields, including social networks (Xiao et al., 2020; Fan et al., 2019; Hao et al., 2024), citation networks (Cummings & Nassar, 2020; Gao et al., 2021), and biological systems (Ktena et al., 2017; Zhang et al., 2021b; Xu et al., 2021; 2020; Bai et al., 2020),

due to its ability to model complex relationships between entities. Traditional node classification approaches in these contexts typically operate under a *closed-set* assumption, where both the labeled and unlabeled data are drawn from the same class distribution. However, in many real-world scenarios, models are expected to operate in an *open-set* environment, encountering previously unseen classes or out-of-distribution (OOD) data during test time (Zhang et al., 2021a; Qin et al., 2024; Dong et al., 2024; Liu et al., 2023; Guo et al., 2023; Fang et al., 2022; Yang et al., 2024a; Heo & Kang, 2024). This discrepancy between training and test distributions necessitates models that can classify in-distribution (ID) data while reliably detecting OOD instances, despite being trained exclusively on ID data. Some recent works (Wu et al., 2020; Song & Wang, 2022; Zhao et al., 2020; Wu et al., 2023) have aimed to extend graph neural networks (GNNs) to the open-set learning paradigm. On the one hand, these approaches enable GNNs to perform robustly under open-set conditions. On the other hand, they assume access to *a large amount of labeled ID data* for training. However, accessing a large amount of labeled data is unrealistic for large-scale, real-world graphs, where obtaining accurate labeled data is costly and time-consuming (Settles, 2009; Tang et al., 2021). Without considering the label budget, current solutions become impractical for resource-constrained settings, where efficient use of labeled data is critical (Liu et al., 2021). Thus, improving the label efficiency is the key to practical graph open-set learning (GOL) methods. As shown in Fig. 1, GOL refers to the task of learning from graph data in scenarios where the unlabeled nodes may belong to unknown classes. The goal is to accurately classify ID (known) nodes while effectively detecting and handling OOD (unknown) nodes. A detailed problem definition of GOL is provided in §3.1.

**Label Efficiency Challenges in GOL.** There are two applicable approaches to addressing this issue. ***First***, label-efficient learning techniques, such as graph active learning (GAL), can effectively reduce labeling costs for closed-set node classification tasks (Cai et al., 2017; Ning et al., 2022; Wu et al., 2019). They focus on selecting the most informative nodes for labeling, minimizing the number of labels required for successful training. Thus, GAL methods prioritize labeling nodes that exhibit high uncertainty or diversity, which often include OOD samples that differ significantly from majority ID samples (Yan et al., 2024; Safaei et al., 2024). However, the literature indicates that open-set learning benefits more from *labeled ID data* than from OOD samples (Ning et al., 2022; Park et al., 2022; Yang et al., 2024b; Safaei et al., 2024), as OOD samples cannot be directly used to train the target ID classifier. When querying OOD examples for labeling, human annotators would disregard these OOD examples as they are unnecessary for the target task, leading to a waste of the labeling budget (Park et al., 2022). Consequently, GOL benefits more from labeled ID samples than OOD samples, which makes applying GAL methods for GOL non-ideal. ***Second***, there are also label-efficient learning for open-set scenarios in the context of image classification Ning et al. (2022); Park et al. (2022); Yang et al. (2024b); Mao et al. (2024); Han et al. (2023); Yan et al. (2024). However, the grid-like structure of image data—where each sample is assumed independent—differs fundamentally from the interconnected nature of graphs. In graph data, nodes are not isolated but influenced by their neighboring nodes, presenting unique challenges. Thus, label-efficient open-set methods for images do not directly translate to GOL, where dependencies between nodes must be considered in the design. In summary, addressing the intersection of open-set classification, label efficiency, and OOD detection for graph data—referred to as GOL in this work—presents a complex challenge. There is a pressing need for novel approaches that balance label efficiency and accurate OOD detection in graphs, ensuring that models generalize effectively while operating within labeling constraints.

**Our Observations and Proposal.** Consider a citation network where the goal is to classify papers into specific ML research fields, such as robotics, computer vision, and natural language processing (see Fig. 1). However, the network also includes papers from unrelated fields like neuroscience and biology, which do not contribute to the primary classification task. Labeling these OOD papers would be inefficient, as they do not aid in training the ID classifier, as we discussed above. Therefore, a practical label-efficient GOL method is needed, with two key properties: (1) accurately identifying and filtering OOD nodes and (2) selecting the most informative ID nodes for labeling to enhance classifier training.

In this work, we focus on label-efficient graph open-set learning (see the problem definition in §3.1), aiming to train an accurate and robust ID classifier within a specific label budget. Our objective is to select informative ID examples for training, enabling the classifier to make confident predictions for ID nodes and effectively detecting OOD nodes, where the classifier is expected to show low confidence. To achieve this, we propose

a label-efficient graph open-set learning framework, termed LEGO-Learn. Our approach addresses two key challenges: (1) filtering out OOD nodes while preserving informative ID nodes, and (2) selecting the most valuable ID nodes for labeling. As shown in Fig. 2, we start by tackling the first challenge: filtering OOD nodes. We introduce a GNN-based filter, which captures graph structure and node dependencies to identify potential OOD nodes. This step ensures that most OOD nodes are removed before any labeling occurs. Next, we focus on selecting the most informative ID nodes for labeling. Given the limited label budget, we use the K-Medoids algorithm to choose diverse, representative nodes from the filtered potential ID set. This ensures that the labeled nodes provide maximum utility for training the classifier by covering various parts of the graph's feature space. Once we have the labeled ID nodes, they are used to train our ID classifier. However, we found that while a strong filter effectively removes OOD nodes, it can also discard informative ID nodes. To mitigate this, we design a $(C + 1)$ class classifier with a weighted cross-entropy loss. This classifier reduces the risk of over-filtering by penalizing the removal of valuable ID nodes, ensuring that the filter retains a balanced set of useful data for further training.

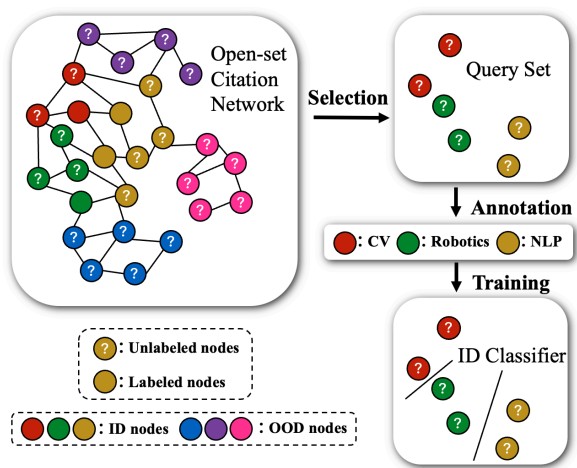

Figure 1: The illustration of graph open-set classification in a citation network. We aim to classify papers into ML research fields such as robotics, CV, and NLP (ID classes). We want to find nodes from ID classes to train the ID classifier.

Our technical contributions include:

- **First Label-efficient Graph Open-set Learning Method**. To the best of our knowledge, we are the first to identify this critical real-world problem and provide a thorough setting and solution.

- **Novel Framework Design**. We introduce LEGO-Learn, a general framework that effectively filters out potential OOD nodes while selecting highly informative ID nodes for manual labeling. The core of LEGO-Learn is a GNN-based filter, designed with a weighted cross-entropy loss to balance between *purity*—the proportion of retained ID examples—and *informativeness*—the usefulness of selected examples for improving the target task.

- **Effectiveness**. LEGO-Learn is validated on four extensive node classification datasets, all under label budget constraints. Experimental results show that LEGO-Learn can efficiently filter out OOD nodes and improve both ID classification and OOD detection. Specifically, LEGO-Learn achieves up to a 6.62% increase in classification accuracy and a 7.49% improvement in AUROC for OOD detection compared to 12 baseline methods. The code is available at: `https://github.com/zhengtaoyao/lego`.

## 2 Related Work

### 2.1 Node-level Graph Open-set Learning

Recent literature has extensively studied the detection of OOD samples on which models should exhibit low confidence. OODGAT (Song & Wang, 2022) introduces a GNN model that explicitly captures the interactions between different kinds of nodes, effectively separating inliers from outliers during feature propagation. OpenWGL (Wu et al., 2020) introduces an uncertain node representation learning method based on a variational graph autoencoder, which can identify nodes from unseen classes. GKDE (Zhao et al., 2020) introduces a network for uncertainty-aware estimation, designed to predict the Dirichlet distribution of nodes and detect OOD data. GNNSafe (Wu et al., 2023) demonstrates that standard GNN classifiers inherently have strong capabilities for detecting OOD samples and introduces a provably effective OOD discriminator built on an energy function derived directly from graph neural networks trained using standard classification

loss. However, all of these methods assume the availability of sufficient ID labels in an open-set scenario, which is not always practical in real-world situations where labeled data is often costly to obtain.

## 2.2 Label-efficient Graph Learning

Many label-efficient learning methods are designed to optimize the performance of semi-supervised node classification under a label budget constraint. AGE (Cai et al., 2017) selects the most informative nodes for training by considering both graph-based information (such as node centrality) and the learned node embeddings (including node classification uncertainty and embedding representativeness). FeatProp (Wu et al., 2019) selects nodes by propagating their features through the graph structure, followed by K-Medoids clustering, which makes it less susceptible to inaccuracies in the representations learned by under-trained models. Recently, based on K-Medoids clustering to select important nodes, MITIGATE (Chang et al., 2024) devises a masked aggregation mechanism for generating distance features that consider representations in latent space and features in the previously labeled set. These graph label-efficient learning methods are usually based on the **closed-set** assumption that the unlabeled data are drawn from known classes, which causes them to fail in open-set node classification tasks. OWGAL (Xu et al., 2023) studies the learning problem on evolving graphs with insufficient labeled data and known classes. It uses prototype learning and label propagation to assign high uncertainty scores to target nodes in both the representation and topology spaces. While also considering label efficiency in open-world settings, their approach differs from ours by dynamically expanding the GNN classifier to accommodate new known classes instead of detecting OOD nodes. In contrast, our goal is to train a robust ID classifier, with a fixed number of known classes, that achieves high label efficiency in the open-set setting.

## 2.3 Label-efficient Open-set Annotation

Label-efficient open-set learning has been widely studied in image classification field. LfOSA (Ning et al., 2022) is one of the first active learning framework for real-world large-scale open-set annotation tasks. It can precisely select examples of known classes by decoupling detection and classification. MQ-Net (Park et al., 2022) utilizes meta-learning techniques to achieve the optimal balance between purity and informativeness. PAL (Yang et al., 2024b) evaluates unlabeled instances based on informativeness and representativeness, balancing pseudo-ID and pseudo-OOD instances in each round. Actively querying pseudo-OOD instances improves both the ID classifier and OOD detector. EOAL (Safaei et al., 2024) quantifies uncertainty using closed-set and distance-based entropy scores to distinguish known from unknown samples, then applies clustering to select the most informative instances. Yan et al. (2024) selects highly informative ID samples by balancing two proposed criteria: contrastive confidence and historical divergence, which measure the possibility of being ID and the hardness of a sample, respectively. However, these approaches are based on the assumption that data samples are generated independently, making them difficult to apply to graph-structured data, where node instances are interdependent.

# 3 Proposed Method

## 3.1 Problem Definition

**Node-level Graph Open-set Classification**. Given a graph $\mathcal{G} = (\mathcal{V}, \mathcal{E}, \mathbf{X})$, where $\mathcal{V}$ is the set of nodes and $\mathcal{E}$ is the set of edges, along with node features $\mathbf{X} = \{x_v \mid v \in \mathcal{V}\}$ and node labels $\mathbf{Y} = \{y_v \mid v \in \mathcal{V}\}$. We have a limited labeled node set $\mathcal{V}_L$ and a large unlabeled node set $\mathcal{V}_U$, where $\mathcal{V}_L = \{v_i^L\}_{i=1}^{n_L}$ and $\mathcal{V}_U = \{v_j^U\}_{j=1}^{n_U}$. Each labeled node $v_i^L$ belongs to one of $C$ known classes $Y_L = \{y_c\}_{c=1}^{C}$, while an unlabeled node $v_j^U$ may belong to an unknown class not included in $Y_L$.

The problem is framed in a semi-supervised, transductive setting, where we can access the full set of nodes during training but only a subset of class labels (ID classes). In general, the task is twofold:

1. **OOD Detection**: For each node $v \in \mathcal{V}$, determine whether it belongs to one of the in-distribution $C$ known classes or from the out-of-distribution unknown classes.

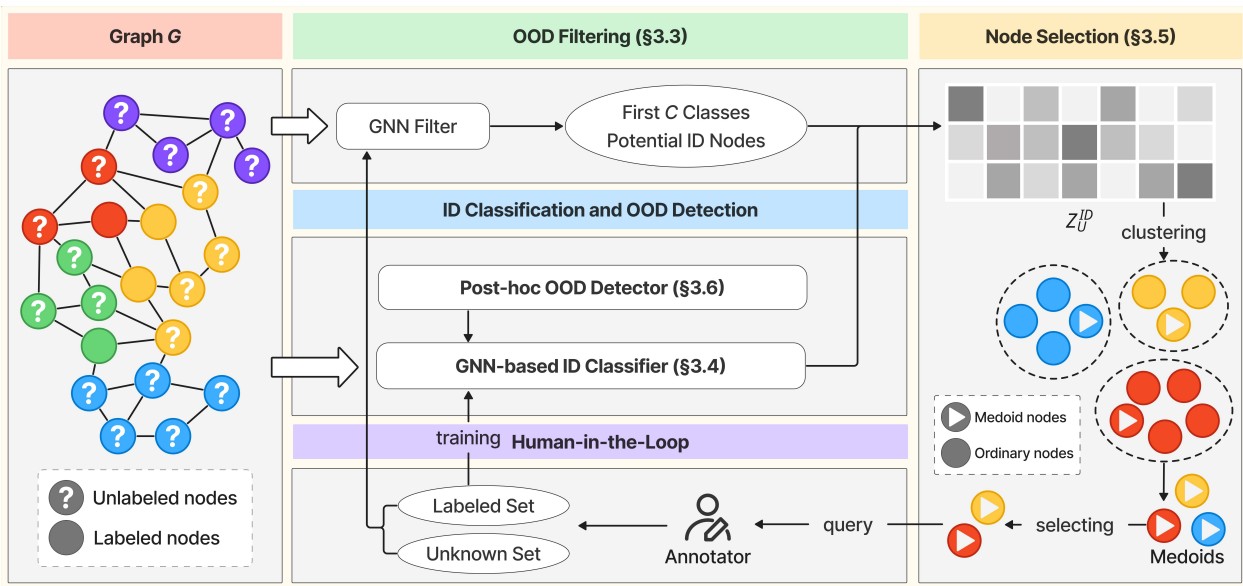

Figure 2: An overview of our framework LEGO-Learn. The first step is to use a GNN-based filter to identify and remove OOD nodes, while using a $C + 1$ classifier with weighted cross-entropy loss to avoid mistakenly eliminating valuable ID nodes (§3.3). A K-Medoids-based node selection method (§3.5) is then applied to choose the most informative ID nodes, which are annotated and used for the next round of training the ID classifier (§3.4). Finally, the filter is retrained with both ID and unknown nodes, and a post-hoc OOD detection method is applied to strengthen the ID classifier's ability to recognize unseen classes (§3.6).

2. **ID Classification**: For nodes identified as ID nodes, classify them into one of the predefined $C$ classes.

**Label-efficient Graph OOD Detection and ID Classification**. Given an initial training set $\mathcal{V}_{\text{train}}$ on $\mathcal{G}$, a labeling budget $\mathcal{B}$, and a loss function $l$, our goal is to selectively construct a query set that contains as many known (ID) examples as possible. At the same time, we aim to select the most informative and representative ID nodes. After selection, the chosen ID nodes are labeled with class labels, while the remaining selected nodes are considered unknown (OOD) nodes.

The goal of the selection procedure is to select a subset of nodes under a label budget constraint, such that querying their labels will allow us to construct a strong ID classifier. Thus we want to select a subset of nodes $\mathcal{V}^s \subset \mathcal{V} \setminus \mathcal{V}_{\text{train}}$ that produces a model $f$ with the lowest loss on the remaining nodes $\mathcal{V}_{\text{test}}$:

$$\arg \min_{\mathcal{V}^s : |\mathcal{V}^s| = \mathcal{B}} \mathbb{E}_{v_i \in \mathcal{V}_{\text{test}}} [l(y_i, \tilde{y}_i)] \tag{1}$$

where $f$ is our target ID classifier, $\tilde{y}_i$ is the label prediction with $f$ of node $v_i$. Specifically, let $\mathcal{V}_L$ be the current labeled set, $\mathcal{V}_U$ the pool of unlabeled nodes available for selection, and $\mathcal{V}_O$ the set of nodes identified as OOD, i.e., not belonging to any known ID class. Then after querying the labels of the selected nodes $\mathcal{V}^s$, $p$ known nodes $\mathcal{V}_L^s$ are annotated and the labeled set is updated to $\mathcal{V}_L = \mathcal{V}_L \cup \mathcal{V}_L^s$, while $q$ nodes $\mathcal{V}_O^s$ with unknown OOD classes are added to the unknown set $\mathcal{V}_O$ and $\mathcal{V}_O = \mathcal{V}_O \cup \mathcal{V}_O^s$. Also, $\mathcal{V}_U = \mathcal{V}_U \setminus \mathcal{V}^s$ and $|\mathcal{B}| = p + q$. Thus, the precision of ID classes can be defined as:

$$\text{precision} = \frac{p}{p + q} \tag{2}$$

## 3.2 Overview of LEGO-Learn

In real-world scenarios, graphs often contain a large number of unlabeled nodes, many of which may be OOD nodes and irrelevant to the target task. Remember that our objective is to train an ID classifier using

only a limited number of ID labels, aiming for high accuracy in ID classification while effectively detecting OOD data, where the classifier is expected to exhibit low confidence. Therefore, we strive to filter out as many OOD nodes as possible before labeling. To achieve this, the *first* step is to design a GNN-based filter to identify and remove potential OOD nodes (see §3.3). However, many useful ID nodes often exhibit high prediction uncertainty, especially during the initial rounds of training. These "hard but informative" ID nodes are more likely to be mistaken for OOD nodes. As a result, filtering out OOD nodes may also unintentionally remove a significant number of these valuable ID nodes. To mitigate this issue, we employ a $C+1$ classifier with a weighted cross-entropy loss to strike a balance between purity—ensuring that filtered nodes are truly OOD—and informativeness—retaining as many useful ID nodes as possible.

Using the current labeled ID nodes, we can train the target ID classifier and generate node representations (see §3.4). Then we can obtain the representation of the unlabeled potential ID nodes from the ID classifier and the filter. Next, a K-Medoids-based node selection method is applied to select the most informative nodes from the unlabeled potential ID nodes (see §3.5). After these informative nodes are annotated, the newly labeled ID nodes can be used in the next round of training for the target ID classifier. Additionally, both the unknown OOD nodes and the labeled ID nodes are employed to retrain the GNN filter. After training the ID classifier with all the annotated nodes, any post-hoc OOD detection methods can be applied to the classifier to enhance its ability to recognize unseen classes (see §3.6). Fig. 2 illustrates the pipeline of the proposed framework LEGO-Learn.

---

**Algorithm 1** The LEGO-Learn algorithm

---

1: **Require:** Current filter $f_\theta$ and classifier $g_\theta$, current labeled set $\mathcal{V}_L$, unknown set $\mathcal{V}_O$ and unlabeled set $\mathcal{V}_U$, query batch size $b$
2: **Ensure:** $\theta_f$, $\theta_g$, $\mathcal{V}_L$, $\mathcal{V}_O$ and $\mathcal{V}_U$ for the next iteration
3: **Process:**
4:    **# Filter training**                                                            ▷ §3.3
5:    Update $\theta_f$ by minimizing $\mathcal{L}_f$ in Eq. (4) using $\mathcal{V}_L$ and $\mathcal{V}_O$
6:    Get the current potential unlabeled ID nodes $\mathcal{V}_U^{ID}$ from the first $C$ classes of prediction of unlabeled set $\mathcal{V}_U$
7:    **# ID classifier training**                                         ▷ §3.4
8:    Update $\theta_g$ by minimizing $\mathcal{L}$ in Eq. (7) using $\mathcal{V}_L$
9:    Get the embeddings $\mathbf{H}_U^{ID}$ of nodes $\mathcal{V}_U^{ID}$
10:   **# K-Medoids based node selection**                              ▷ §3.5
11:   Compute pairwise distance based on embeddings $\mathbf{H}_U^{ID}$ and get $m$ medoids
12:   Select $b$ nodes $\mathcal{V}^s$ from the $m$ medoids with the highest uncertainty of prediction
13:   **# Node annotation**
14:   Query the selected nodes' labels and obtain $\mathcal{V}_L^s$ and $\mathcal{V}_O^s$, where $|\mathcal{V}_L^s| + |\mathcal{V}_O^s| = |\mathcal{V}^s| = b$
15:   Update labeled, unknown, and unlabeled sets: $\mathcal{V}_L = \mathcal{V}_L \cup \mathcal{V}_L^s$, $\mathcal{V}_O = \mathcal{V}_O \cup \mathcal{V}_O^s$ and $\mathcal{V}_U = \mathcal{V}_U \setminus \mathcal{V}^s$
16: **Output:** $\theta_f$, $\theta_g$, $\mathcal{V}_L$, $\mathcal{V}_O$ and $\mathcal{V}_U$ for the next iteration
17: **# OOD detection**                                                            ▷ §3.6
18: Apply post-hoc OOD detection methods to the trained ID classifier for identifying OOD nodes.

---

## 3.3 Step 1: Removing OOD Nodes via GNN Filter

Assume that we want to train a C-class ID classifier. To achieve this, we extend the filter with an additional $(C+1)-th$ output to predict the unknown class. We want to detect whether a node belongs to an unknown $(C+1)-th$ class, possibly filtering out as many OOD nodes as possible before annotation. In this case, we select nodes predicted to belong to the first $C$ known classes for the next step, while excluding those identified as unknown. In this paper, we use a two-layer standard graph convolutional network (GCN) as the OOD filter, and set the output dimension of the last layer as $C+1$.

The output of the OOD filter is the representation matrix $\mathbf{Z}_{filter} \in \mathbb{R}^{N \times (C+1)}$ for all nodes:

$$\mathbf{Z}_{filter} = GNN_{filter}(\mathbf{A}, \mathbf{X}) \tag{3}$$

Here, we denote the OOD filter as $GNN_{filter}$. However, sometimes the graph has many OOD nodes (i.e, the number of OOD nodes is much larger than the number of nodes of any ID class). To prevent filtering out too many useful ID nodes, we use the following weighted cross-entropy loss function:

$$\mathcal{L}_f = -\sum_{i=1}^{N}(\sum_{c=1}^{C} y_i^{(c)} \log \hat{p}_i^{(c)} + w^{(C+1)} y_i^{(C+1)} \log \hat{p}_i^{(C+1)}). \tag{4}$$

Here we introduce the weight $w^{(C+1)}$ to balance the purity and informativeness. While a larger $w^{(C+1)}$ can effectively filter out OOD nodes, it may also unintentionally remove many highly informative ID nodes, which is undesirable for the later process. Conversely, a smaller $w^{(C+1)}$ is preferable when OOD nodes dominate the training set, as it prevents the excessive removal of valuable ID data.

With $Z_{filter}$ from the GNN filter, we can select the nodes predicted to belong to the first $C$ classes as potential ID nodes and pass them to the node selection module to further select the most informative ones for labeling. The unlabeled set is denoted as $\mathcal{V}_U$, and the potential ID node set is represented as $\mathcal{V}_U^{ID}$.

To specify the relationship between $\mathcal{V}_U^{ID}$ and $\mathcal{V}_U$ more precisely, we use the output of the OOD filter $\mathbf{Z}_{filter}$ to identify the indices of the potential unlabeled ID nodes by selecting nodes whose predicted class labels fall within the known classes 1 to $C$:

$$idx = \left\{ i \in \mathcal{V}_U \mid \arg\max_c \mathbf{Z}_{filter}[i,c] \leq C \right\} \tag{5}$$

Finally, the potential ID node set is defined as:

$$\mathcal{V}_U^{ID} = \left\{ v_i \mid i \in idx \right\} \tag{6}$$

### 3.4 Step 2: Train ID Classifier with the Current ID Node Set

With the help of the OOD filter, we can train the target ID classifier with more labeled ID nodes while adhering to the label budget constraint. Assume that the current labelled node set is $\mathcal{V}_L$, then the cross-entropy loss function for node classification over the labeled node set is defined as:

$$\mathcal{L} = -\frac{1}{|\mathcal{V}_L|} \sum_{i \in \mathcal{V}_L} \sum_{c=1}^{C} y_{ic} \ln \hat{y}_{ic} \tag{7}$$

Here, any GNN can be used as the ID classifier. For example, consider a two-layer GCN. The output of the first layer is as follows:

$$\mathbf{H} = \sigma\left(\tilde{\mathbf{D}}^{-\frac{1}{2}} \tilde{\mathbf{A}} \tilde{\mathbf{D}}^{-\frac{1}{2}} \mathbf{X} \mathbf{W}^{(0)}\right) \tag{8}$$

where $\tilde{\mathbf{A}} = \mathbf{A} + \mathbf{I}$ and $\tilde{\mathbf{D}}_{ii} = \sum_j \tilde{\mathbf{A}}_{ij}$, $\mathbf{I}$ is the identity matrix, and $\mathbf{W}^{(0)}$ is the weight matrix. This propagated node feature matrix $\mathbf{H}$ is used for subsequent clustering to select the most informative nodes. The final output of the ID classifier is the representation matrix $\mathbf{Z} \in \mathbb{R}^{N \times C}$ for all nodes, which is:

$$\mathbf{Z} = \sigma\left(\tilde{\mathbf{D}}^{-\frac{1}{2}} \tilde{\mathbf{A}} \tilde{\mathbf{D}}^{-\frac{1}{2}} \mathbf{H} \mathbf{W}^{(1)}\right) \tag{9}$$

### 3.5 Step 3: K-Medoids Based Node Selection

Most node selection methods generally prioritize nodes with high prediction uncertainty or diverse representations for labeling. However, open-set noise distorts these metrics, as OOD nodes also exhibit high uncertainty and diversity due to their lack of class-specific features or shared inductive biases with ID examples. Thanks to our OOD filter, which removes many OOD nodes, we can focus on selecting the most informative ones from the remaining potential ID nodes.

After obtaining the indices $idx$ of the first $C$ classes of unlabeled nodes predicted by the OOD filter and the propagated node feature matrix $\mathbf{H}$ from the ID classifier, we can derive the representation of potential unlabeled ID nodes:

$$\mathbf{H}_U^{ID} = \mathbf{H}(idx, :) \tag{10}$$

Then we compute the pairwise Euclidean distance between nodes in the unlabeled potential ID node set:

$$d(v_i, v_j) = \|h_i - h_j\|_2, \tag{11}$$

where $h_i$ and $h_j$ are the propagated node feature of node $v_i$, $v_j$ from the ID classifier. Similar to prior studies (Liu et al., 2022), we then perform K-Medoids clustering on $\mathcal{V}_U^{ID}$, where the centers selected for the candidate set must be actual nodes within the graph. The number of clusters is defined as m. After getting the m medoids, we select the top $b$ nodes with the highest uncertainty of prediction for annotation.

### 3.6 Post-hoc OOD Detection

Our goal is to develop a powerful and robust ID classifier. To achieve this, the classifier must excel in two key aspects: first, it should provide accurate and high-confidence predictions for ID nodes; second, it should give low-confidence predictions for OOD nodes, ensuring effective differentiation between the two. In this way even without a post-hoc method the ID classifier inherently possesses the ability to identify an OOD node.

Any post-hoc OOD detectors (Liang et al., 2017; Lee et al., 2018; Hendrycks & Gimpel, 2016; Yang et al., 2022) can be applied into our framework, since the ID classifier is already capable of OOD detection (Wu et al., 2023; Vaze et al., 2021), and a good post-hoc method will strengthen its OOD detection ability. For each node $v_i$, it has a final representation vector $z_i \in \mathbb{R}^C$ from the output of ID classifier. Here we simply use the entropy of the predicted class distribution as OOD scores as in Macêdo et al. (2021); Ren et al. (2019). The entropy of $z_i$ is calculated as follows:

$$e_i = -\sum_{j=1}^{C} z_{ij} \log(z_{ij}) \tag{12}$$

The higher the entropy of $z_i$, the more likely it is that node $v_i$ belongs to an unknown class. The process of LEGO-Learn approach is summarized in Algorithm 1.

## 4 Experiments

Our experiments address the following research questions (RQ): RQ1 (§4.2): How effective is the proposed LEGO-Learn in ID class classification and node-level OOD detection in comparison to other leading baselines? RQ2 (§4.3 and §4.4): What nodes are important for open-set classification and how does the OOD filter influence the open-set learning performance? RQ3 (§4.5): How do different design modules in LEGO-Learn impact its effectiveness?

### 4.1 Experimental Setup

#### 4.1.1 Datasets

We test LEGO-Learn on four real-world datasets (Sen et al., 2008; Shchur et al., 2018; McAuley et al., 2015) that are widely used as benchmarks for node classification, i.e., Cora, AmazonComputers, AmazonPhoto and LastFMAsia. We preprocess datasets using the same pipeline as in Song & Wang (2022). For each dataset, we split all classes into ID and OOD sets, ensuring that the ID classes are relatively balanced in terms of node count. The OOD class and OOD ratio for the four datasets are shown in Appendix A. Additionally, the number of ID classes is set to a minimum of three to prevent overly simple classification tasks.

For each dataset with $C$ ID classes, we randomly select $10 \times C$ of ID nodes and the same number of OOD nodes as the validation set. We then randomly select 500 ID nodes and 500 OOD nodes as the test set. All remaining nodes constitute the "unlabeled node pool".

Table 1: Performance comparison (best highlighted in bold) of different models on ID classification and OOD detection tasks for Cora and Amazon-CS datasets. LEGO-Learn achieves the best across all baselines.

| Model | Method | Cora | | | | Amazon-CS | | | |
|---|---|---|---|---|---|---|---|---|---|
| | | ID ACC ↑ | AUROC ↑ | AUPR ↑ | FPR ↓ | ID ACC ↑ | AUROC↑ | AUPR ↑ | FPR↓ |
| GCN-ENT | Random | 0.8254 | 0.7742 | 0.7821 | 0.5190 | 0.7420 | 0.6686 | 0.6833 | 0.6604 |
| | Uncertainty | 0.8112 | 0.7609 | 0.7585 | 0.5689 | 0.5920 | 0.5963 | 0.5952 | 0.7788 |
| | FeatProp | 0.8530 | 0.7678 | 0.7696 | 0.5500 | 0.7346 | 0.7082 | 0.7121 | 0.6079 |
| | MITIGATE | 0.8346 | 0.7551 | 0.7649 | 0.5511 | 0.6304 | 0.6666 | 0.6551 | 0.7039 |
| OODGAT | Random | 0.7287 | 0.7861 | 0.8165 | 0.4533 | 0.8048 | 0.7533 | 0.7652 | 0.4849 |
| | Uncertainty | 0.7884 | 0.7961 | 0.8173 | 0.4615 | 0.7428 | 0.7564 | 0.7853 | 0.4630 |
| | FeatProp | 0.7772 | 0.8055 | 0.8233 | 0.4414 | 0.8044 | 0.8135 | 0.8276 | 0.3757 |
| | MITIGATE | 0.7998 | 0.7965 | 0.8175 | 0.4579 | 0.7950 | 0.8446 | 0.8361 | 0.3553 |
| GNNSafe | Random | 0.8166 | 0.7390 | 0.7585 | 0.7092 | 0.7490 | 0.6445 | 0.6437 | 0.7836 |
| | Uncertainty | 0.7892 | 0.7226 | 0.7163 | 0.6582 | 0.6656 | 0.6031 | 0.5982 | 0.7864 |
| | FeatProp | 0.8448 | 0.7473 | 0.7560 | 0.6640 | 0.7290 | 0.6820 | 0.6620 | 0.7034 |
| | MITIGATE | 0.8410 | 0.7633 | 0.7728 | 0.6256 | 0.7088 | 0.6536 | 0.6390 | 0.7440 |
| LEGO-Learn | | **0.8684** | **0.8804** | **0.8878** | **0.2869** | **0.8710** | **0.8533** | **0.8474** | **0.3216** |

Table 2: Performance comparison (best highlighted in bold) of different models on ID classification and OOD detection tasks for Amazon-photo and LastFMAsia datasets. LEGO-Learn achieves the best across all baselines.

| Model | Method | Amazon-photo | | | | LastFMAsia | | | |
|---|---|---|---|---|---|---|---|---|---|
| | | ID ACC ↑ | AUROC ↑ | AUPR ↑ | FPR ↓ | ID ACC ↑ | AUROC ↑ | AUPR ↑ | FPR ↓ |
| GCN-ENT | Random | 0.9054 | 0.7970 | 0.7937 | 0.4883 | 0.6902 | 0.7990 | 0.8148 | 0.4702 |
| | Uncertainty | 0.7784 | 0.7228 | 0.7158 | 0.6270 | 0.6170 | 0.7306 | 0.7366 | 0.6079 |
| | FeatProp | 0.8950 | 0.8014 | 0.7983 | 0.4888 | 0.6518 | 0.7765 | 0.7788 | 0.5425 |
| | MITIGATE | 0.8552 | 0.7818 | 0.7744 | 0.5234 | 0.6598 | 0.7719 | 0.7751 | 0.5547 |
| OODGAT | Random | 0.9306 | 0.8776 | 0.8891 | 0.2863 | 0.7616 | 0.8752 | 0.8917 | 0.3005 |
| | Uncertainty | 0.9098 | 0.8173 | 0.8336 | 0.3978 | 0.6888 | 0.8307 | 0.8518 | 0.3963 |
| | FeatProp | 0.9390 | 0.8823 | 0.8806 | 0.2955 | 0.7278 | 0.8662 | 0.8786 | 0.3406 |
| | MITIGATE | 0.9264 | 0.8628 | 0.8693 | 0.3373 | 0.7088 | 0.8448 | 0.8645 | 0.3735 |
| GNNSafe | Random | 0.8936 | 0.7664 | 0.7560 | 0.5869 | 0.6980 | 0.7885 | 0.7998 | 0.5997 |
| | Uncertainty | 0.8066 | 0.7077 | 0.6970 | 0.6873 | 0.6470 | 0.6962 | 0.6905 | 0.6957 |
| | FeatProp | 0.8794 | 0.7550 | 0.7297 | 0.5931 | 0.6652 | 0.7318 | 0.7240 | 0.6527 |
| | MITIGATE | 0.8672 | 0.7695 | 0.7542 | 0.5944 | 0.6662 | 0.7113 | 0.7046 | 0.6868 |
| LEGO-Learn | | **0.9648** | **0.9257** | **0.9279** | **0.1856** | **0.7818** | **0.8852** | **0.9035** | **0.2627** |

### 4.1.2 Baselines

We compare LEGO-Learn with two types of baselines: (1) graph OOD detection methods, including GCN-ENT (Kipf & Welling, 2016), OODGAT (Song & Wang, 2022), GNNSafe (Wu et al., 2023) (2) node selection methods for node classification, including random, uncertainty (Luo et al., 2013), FeatProp (Wu et al., 2019), MITIGATE (Chang et al., 2024). However, since there are no current methods specifically designed for our setting, we need to modify the baselines to better fit our context. In general, we use the following baselines:

- GCN-ENT-random, GCN-ENT-uncertainty, GCN-ENT-FeatProp, GCN-ENT-MITIGATE: we use GCN as the backbone of ID classifier and node embeddings' entropy as post-hoc OOD detection method. We use four different selection methods to select training nodes to label: (1) randomly select nodes from the unlabeled node pool (2) select the nodes with largest uncertainty of predictions (3) use graph active learning method FeatProp (Wu et al., 2019) to select nodes (4) use the K-Medoids based method with masked aggregation mechanism proposed in Chang et al. (2024) to select nodes.

- OODGAT-random, OODGAT-uncertainty, OODGAT-FeatProp, OODGAT-MITIGATE: we use OODGAT as the ID classification and OOD detection backbone and combine it with the previous four selection strategies.

- GNNSafe-random, GNNSafe-uncertainty, GNNSafe-FeatProp, GNNSafe-MITIGATE: similarly, we use GNNSafe as the graph open-set classification backbone, combining it with the previous four selection strategies.

For our model LEGO-Learn, we use two OODGAT layers as ID classifier and entropy as OOD score. We use two standard GCN layers as the OOD filter. It is important to note that we rely solely on the ID classifier for OOD detection, as our goal is to assess its ability to make accurate predictions for ID nodes while effectively identifying OOD nodes.

### 4.1.3 Evaluation Metrics

For the ID classification task, we use classification accuracy (ID ACC) as the evaluation metric. For the OOD detection task, we employ three commonly used metrics from the OOD detection literature (Song & Wang, 2022): the area under the ROC curve (AUROC), the precision-recall curve (AUPR), and the false positive rate when the true positive rate reaches 80% (FPR@80). In all experiments, OOD nodes are considered positive cases. Details about these metrics are provided in Appendix B.

### 4.1.4 Implementation Details

The initial label budget for all datasets is 5 nodes per ID class. The total label budget is 15 nodes per ID class. For each round of selection, we select $2 \times C$ of nodes from the unlabeled pool and annotate the selected nodes for all methods. It should be noted that although the total number of final labeled nodes is $15 \times C$, many of these labeled nodes may be OOD nodes and therefore cannot be used to train the ID classifier.

For all K-Medoids based selection methods, the number of clusters is set to 48. In each round of node selection, we select $2 \times C$ of nodes with the highest uncertainty of prediction from the 48 medoids. All GCNs have 2 layers with hidden dimensions of 32. The weight for the unknown class in the filter's loss function is chosen from $\{0.001, 0.1, 0.2\}$ based on the results of the validation set. All models use a learning rate of 0.01 and a dropout probability of 0.5. We average all results across 10 different random seeds.

### 4.2 Main Results

We present the performance comparison of different models on ID classification and OOD detection tasks for four datasets in Tables 1 and 2. From the results, we make the following observations:

1. **LEGO-Learn outperforms all baselines**: Across all datasets, our proposed method LEGO-Learn consistently outperforms all baseline models in both ID classification accuracy and OOD detection metrics. Specifically, on four datasets, for ID classification, the best improvement is observed on Amazon-photo dataset, where ID accuracy increases from 80.48% (achieved by the best baseline OODGAT with random selection) to 87.10%, a 6.62% improvement. Moreover, LEGO-Learn demonstrates remarkable improvement in OOD detection metrics, achieving higher AUROC and AUPR scores while maintaining a lower FPR@80 across all datasets. The best improvement is seen on the Cora dataset, where the OOD AUROC increases from 80.55% (achieved by the best baseline, OODGAT with FeatProp) to 88.04%, a 7.49% improvement.

2. **Effectiveness of K-Medoids clustering**: The baselines incorporating FeatProp and MITIGATE, which use clustering-based node selection methods, generally perform better than those using uncertainty-based selection. This suggests that clustering helps select more informative nodes. However, they still fall short compared to LEGO-Learn, highlighting the effectiveness of our integrated approach that combines OOD filtering with clustering.

3. **Limitations of uncertainty-based selection**: Baselines using uncertainty-based node selection often perform worse than those using random selection because they tend to select nodes with high prediction

uncertainty, which are often OOD nodes in an open-set setting, leading to lower precision and negatively affecting the performance of the ID classifier.

4. **OODGAT layer can enhance OOD detection performance more than ID classification accuracy**: Using the same node selection method, models employing OODGAT layers as the backbone for the ID classifier generally demonstrate significantly better OOD detection performance compared to those with GCN layers. This improvement arises because OODGAT layers are specifically designed for node-level OOD detection, effectively separating ID nodes from OOD nodes during feature propagation. However, due to having more parameters than standard GCN layers, OODGAT layers may struggle to achieve high ID classification accuracy when the number of labeled nodes available for training is limited.

## 4.3 What Nodes Are Important for Graph Open-set Classification?

We calculate the final precision of ID nodes for various selection methods, as defined in Ning et al. (2022). Precision here refers to the ratio of true ID nodes to the total number of selected and annotated nodes. The baseline results are averaged across three different graph open-set learning models. As shown in Table 3, LEGO-Learn achieves the highest precision across all datasets. This demonstrates our method can effectively filter out OOD nodes before node selection and annotation. Interestingly, most node selection methods achieve lower precision compared with random selection. That is because current selection methods tend to select nodes with higher uncertainty in predictions or greater diverse in representations, while nodes with higher uncertainty are more likely to be OOD nodes.

Table 3: Precision values of ID nodes.

| Model | Method | Cora | Amazon-CS | Amazon-Photo | LastFM-Asia |
|---|---|---|---|---|---|
| GCN-ENT | random | 0.4783 | 0.4933 | 0.4853 | 0.4600 |
| | uncertainty | 0.3500 | 0.4053 | 0.2960 | 0.2874 |
| | FeatProp | 0.3383 | 0.4067 | 0.2920 | 0.2622 |
| | MITIGATE | 0.3333 | 0.4080 | 0.2747 | 0.2852 |
| OODGAT | random | 0.4783 | 0.4933 | 0.4853 | 0.4600 |
| | uncertainty | 0.3400 | 0.2973 | 0.3093 | 0.2370 |
| | FeatProp | 0.3350 | 0.3427 | 0.2600 | 0.2444 |
| | MITIGATE | 0.3433 | 0.3040 | 0.2600 | 0.2274 |
| GNNSafe | random | 0.4783 | 0.4933 | 0.4853 | 0.4600 |
| | uncertainty | 0.3467 | 0.4560 | 0.3160 | 0.3274 |
| | FeatProp | 0.3650 | 0.4240 | 0.3013 | 0.2889 |
| | MITIGATE | 0.3667 | 0.4346 | 0.2893 | 0.2889 |
| LEGO-Learn | | **0.5733** | **0.6333** | **0.6173** | **0.6978** |

Although current node selection methods cannot select a high proportion of ID nodes from the unlabeled node pool, they can achieve competitive performance compared to random selection, which has higher precision. This suggests that highly informative nodes are very useful for training the ID classifier. Our method can not only select informative nodes but can also filter out many potential OOD nodes, enabling it to achieve both high precision in ID node selection and high classification accuracy.

## 4.4 A Strong Filter Is Not All You Need

In general, if the GNN filter can remove more OOD nodes, the precision of ID nodes will improve. As a result, the ID classifier will be better trained with more ID nodes and perform more effectively. However, this is not always the case. While a filter with a large $w^{(C+1)}$ (strong filter) can filter out more OOD nodes, it does not always lead to improved performance of the ID classifier. We compare the performance of our method using a strong filter ($w^{(C+1)} = 1$) and a normal filter ($w^{(C+1)} = 0.1$) on LastFMAsia dataset. The results are shown in Table 4.

Table 4: Comparison of strong filter and normal filter performance on LastFMAsia. The strong filter may improve ID classification precision while decreasing the accuracy.

|  | ID ACC ↑ | AUROC ↑ | AUPR ↑ | FPR ↓ | Precision ↑ |
|---|---|---|---|---|---|
| **Strong Filter** | 0.7602 | 0.8818 | 0.9010 | 0.2636 | 0.7444 |
| **Normal Filter** | 0.7810 | 0.8710 | 0.8881 | 0.3036 | 0.6385 |

As we can see from the results, the strong filter significantly increases the precision of ID nodes (from 0.6385 to 0.7444) but does not enhance ID classification performance (a drop from 0.7810 to 0.7602) compared to the normal filter with a smaller $w^{(C+1)}$. This indicates that while a strong filter removes out more OOD nodes before annotation, it also filters out many useful ID nodes. As a result, it becomes impossible to select these valuable nodes in later stages. Often, these informative ID nodes are more similar to OOD nodes due to their high prediction uncertainty. More importantly, this demonstrates that highly informative nodes are crucial for training the ID classifier. Sometimes, using fewer informative nodes is more effective for training the ID classifier than relying on a larger number of simpler nodes. In summary, this suggests that we should use the weight $w^{(C+1)}$ to balance purity and informativeness. We leave the design of a more powerful filter, capable of removing most OOD nodes while retaining the majority of informative ID nodes, for future work.

## 4.5 Ablation Study

To understand the contribution of each component in our proposed framework LEGO-Learn, we conduct an ablation study on Cora dataset by systematically removing or altering specific modules and observing the impact on the performance. Table 5 presents the results of this study across four key variants of our model:

Table 5: Ablation study on Cora. LEGO-Learn achieves the best performance.

|  | ID ACC ↑ | AUROC ↑ | AUPR ↑ | FPR ↓ |
|---|---|---|---|---|
| **LEGO-Learn-ATT** | 0.8644 | 0.8285 | 0.8299 | 0.4220 |
| **LEGO-Learn-Filter** | 0.7998 | 0.7965 | 0.8175 | 0.4579 |
| **LEGO-Learn-Cluster** | 0.7764 | 0.7689 | 0.8023 | 0.4683 |
| **LEGO-Learn** | **0.8684** | **0.8804** | **0.8878** | **0.2869** |

- **LEGO-Learn-ATT**: The OODGAT layer in the GNN-based ID classifier for LEGO-Learn is replaced with GCN-ENT.

- **LEGO-Learn-Filter**: The backbone uses K-Medoids clustering for node selection but omits the OOD filtering process. All unlabeled nodes in the original graph are used for clustering, and the medoids with highest uncertainty are selected for annotation. The annotated ID nodes are used to train the ID classifier.

- **LEGO-Learn-Cluster**: The node clustering part is removed. The backbone employs the GNN-based OOD filter without additional clustering or selection steps. Instead, after filtering, nodes are randomly selected for annotation to train the ID classifier.

**LEGO-Learn vs. LEGO-Learn-Filter/LEGO-Learn-Cluster**: The performance of LEGO-Learn (the complete framework) significantly outperforms both LEGO-Learn-Filter and LEGO-Learn-Cluster, demonstrating the necessity of each component. The node selection module better selects the most informative nodes, allowing the ID classifier to be well-trained. Additionally, the filter is crucial because it removes most of the OOD nodes, thereby improving the purity of ID nodes.

**Impact of OODGAT Layer (LEGO-Learn-ATT)**: Replacing the original OODGAT layer with a standard GCN-ENT model (LEGO-Learn-ATT) results in a noticeable drop in both AUROC (from 0.8804 to 0.8285) and AUPR (from 0.8878 to 0.8299). The GNN-based OODGAT layer clearly enhances the performance of the model in detecting OOD nodes compared to GCN layers, suggesting that the attention mechanism used for separating ID nodes and OOD nodes can lead to better OOD detection.

# 5 Conclusion and Future Directions

In this paper, we addressed the problem of label-efficient graph open-set learning for the first time and introduced LEGO-Learn (Label-Efficient Graph Open-set Learning), a novel framework designed to tackle the challenges of open-set node classification on graphs under strict label budget constraints. LEGO-Learn integrates a GNN-based OOD filter with a K-Medoids-based node selection strategy, enhanced by a weighted cross-entropy loss to balance the retention of informative ID nodes while filtering out OOD nodes. Our extensive experiments on four real-world datasets demonstrate that LEGO-Learn consistently outperforms state-of-the-art methods in both ID classification accuracy and OOD detection under label budget constraints. Ablation studies further highlight the importance of each component in our framework. Future work includes scaling LEGO-Learn to accommodate larger and more complex graphs, developing improved node selection strategies for OOD detection, optimizing the trade-off between purity and informativeness, exploring the performance of graph open-set learning under varying proportions of OOD nodes, and modeling different annotators' accuracy when labeling different types of nodes.

## Acknowledgments

This work was partially supported by the National Science Foundation under Award No. 2428039, No. 2346158, and No. 2106758. Any opinions, findings, and conclusions or recommendations expressed are those of the authors and do not necessarily reflect the views of the National Science Foundation.

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

# A   Datasets Information

Cora is a citation network where each node represents a published paper, and each edge indicates a citation between papers. The objective is to predict the topic of each paper as its label class. This dataset includes 2708 nodes, 10556 edges, 1433 features, and 7 distinct classes.

Amazon-CS is a product co-purchasing network from the Amazon platform, where each node represents a product, and each edge indicates that two connected products are frequently bought together. The label of each node corresponds to the product's category. This dataset contains 13752 nodes, 491722 edges, 767 features, and 10 classes.

Amazon-Photo is a product co-purchasing network on Amazon, where each node represents a product and each edge indicates that two connected products are frequently bought together. The label of each node corresponds to the product's category. This dataset includes 7650 nodes, 238162 edges, 745 features, and 8 classes.

The LastFMAsia dataset consists of 7624 users from the LastFM music platform, who are classified into one of 18 regions based on their listening habits. The graph is built on the basis of users' friend connections, containing 55612 edges. Each user is described by a feature vector of 128 attributes representing the artists they listen to. The task is to classify users into one of the 18 regions based on their musical preferences and social connections.

Table 6: OOD class and OOD ratio for different datasets

| Dataset | OOD class | OOD ratio |
|---|---|---|
| Cora | [0, 1, 3] | 0.51 |
| Amazon-Computer | [0, 3, 4, 5, 9] | 0.49 |
| Amazon-Photo | [1, 6, 7] | 0.52 |
| LastFMAsia | [1, 2, 3, 4, 5, 9, 10, 12, 17] | 0.53 |

Table 7: Statistics for main datasets

| Dataset | #Nodes | #Edges | #Features | #Classes |
|---|---|---|---|---|
| Cora | 2708 | 10556 | 1433 | 7 |
| Amazon-Computer | 13752 | 491722 | 767 | 10 |
| Amazon-Photo | 7650 | 238162 | 745 | 8 |
| LastFMAsia | 7624 | 55612 | 128 | 18 |

# B   Descriptions of Evaluation Metrics

**AUROC** stands for "Area Under the Receiver Operating Characteristic Curve." It is a performance metric used in binary classification tasks to evaluate how well a model distinguishes between positive and negative classes. A higher AUROC value indicates better model performance in distinguishing between the two classes.

**AUPR** stands for the area under the precision-recall (PR) curve, similar to AUC, but it provides a more effective performance evaluation for imbalanced data.

**FPR80** refers to the false positive rate (FPR) when the true positive rate (TPR) reaches 80%. It is used to measure the likelihood that an OOD example is incorrectly classified as ID when most ID samples are correctly identified. A lower FPR80 value indicates better detection performance.

## C   Different Label Budgets

Here, we present the results for different label budgets on LastFMAsia. As shown in Table 8 and Table 9, LEGO-Learn achieves better performance across various label budgets.

Table 8: $20 \times C$ label budget

|  | ID ACC ↑ | AUROC ↑ | AUPR ↑ | FPR80 ↓ |
|---|---|---|---|---|
| GCN-ENT | 0.6844 | 0.7863 | 0.7923 | 0.5158 |
| LEGO-Learn | 0.7826 | 0.9057 | 0.9179 | 0.2287 |

Table 9: $10 \times C$ label budget

|  | ID ACC ↑ | AUROC ↑ | AUPR ↑ | FPR80 ↓ |
|---|---|---|---|---|
| GCN-ENT | 0.6248 | 0.7527 | 0.7518 | 0.5903 |
| LEGO-Learn | 0.7512 | 0.8624 | 0.8812 | 0.3116 |

## D   Standard Deviation Results

We assess the model's stability by presenting the standard deviation values of LEGO-Learn and GCN-ENT for two datasets in Table 10. As shown in the table, our method and the baseline generally exhibit similar variance values.

Table 10: Our method and baseline generally have similar standard deviation values.

|  | ID ACC ↑ | AUROC ↑ | AUPR ↑ | FPR80 ↓ |
|---|---|---|---|---|
| Cora (LEGO-Learn) | 6.54% | 1.40% | 0.30% | 13.63% |
| LastFMAsia (LEGO-Learn) | 3.41% | 3.13% | 2.45% | 10.24% |
| Cora (GCN-ENT) | 6.31% | 5.47% | 5.64% | 14.93% |
| LastFMAsia (GCN-ENT) | 4.25% | 2.90% | 3.32% | 12.63% |

## E   OOD Detection Under Different Category Divisions

To demonstrate generalizability, we modify the OOD divisions of Cora (where the ID classes are now 1, 2, 3, and 4) and LastFMAsia (where the ID classes range from 0 to 8). The results of LEGO-Learn and GCN-ENT, using the same K-Medoids method, are presented in Table 11. From these results, we observe that LEGO-Learn consistently outperforms the baseline, regardless of the division.

## F   Parameter Sensitivity Analysis

The ID classification accuracy of LEGO-Learn under different $w$ values in the filter for LastFMAsia is presented in Table 12.

## G   OOD Scores

We visualize the OOD scores predicted by LEGO-Learn for ID and OOD nodes in the test set. As shown in the Fig. G, the scores for ID and OOD inputs are well-separated. This proves that our method effectively identifies OOD nodes and accurately predicts ID nodes.

Table 11: The results of LEGO-Learn and GCN-ENT for another OOD division of Cora and LastFMAsia.

| Model | ID ACC | AUROC | AUPR | FPR80 |
|---|---|---|---|---|
| GCN-ENT (Cora) | 0.8249 | 0.8290 | 0.8216 | 0.4550 |
| LEGO-Learn (Cora) | 0.8376 | 0.9210 | 0.9244 | 0.2086 |
| GCN-ENT (LastFMAsia) | 0.7294 | 0.7575 | 0.7473 | 0.5850 |
| LEGO-Learn (LastFMAsia) | 0.8024 | 0.8684 | 0.8714 | 0.3427 |

Table 12: ID classification accuracy of our method under different $w$ values for LastFMAsia.

| w | 0.01 | 0.1 | 0.2 | 0.3 | 0.4 | 1 |
|---|---|---|---|---|---|---|
| LastFMAsia | 0.7624 | 0.7800 | 0.7770 | 0.7692 | 0.7650 | 0.7602 |

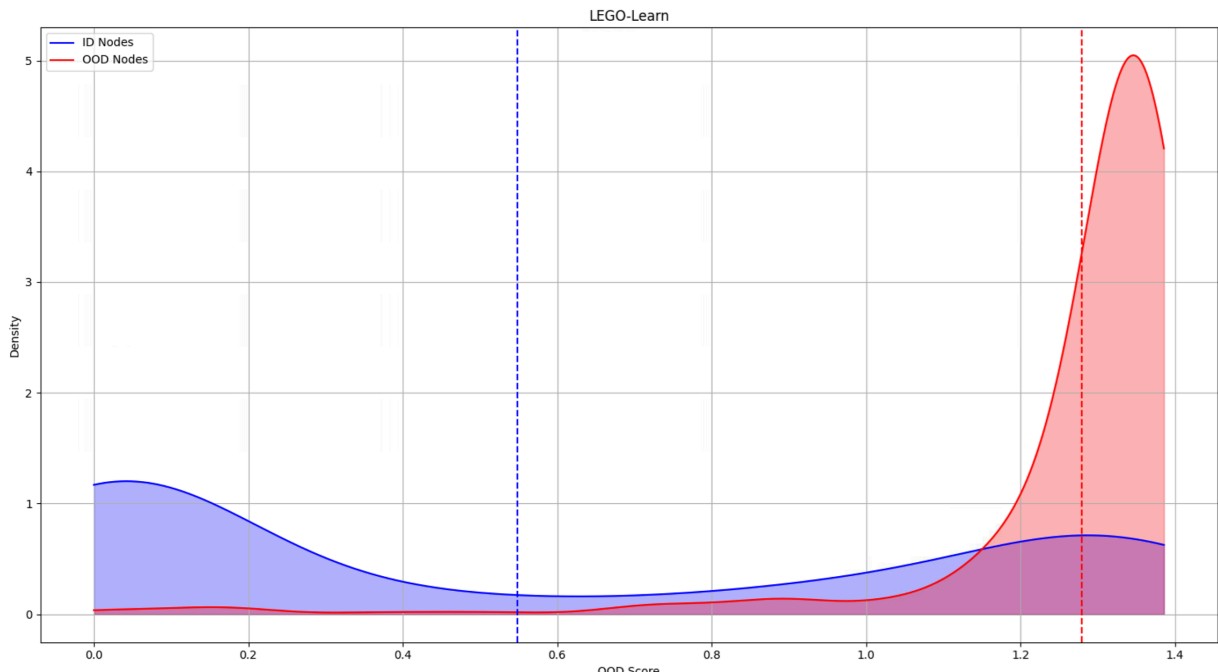

Figure 3: OOD scores histogram

