# OpenReview forum: "LEGO-Learn: Label-Efficient Graph Open-Set Learning"
_TMLR — Accepted by TMLR_

### Review · Reviewer_YY2z · 2025-03-10

**Summary Of Contributions:**

The authors consider the problem of node graph classification, where we only care about (and have data for) a subset of the classes. Can we detect which of the nodes are from a class for which we have no data? This is the problem of open-set learning. The authors combine this setting with active learning to identify what nodes to label (or mark OOD).
Their solution trains a GNN that identifies (potential) OOD (outside class) nodes, and uses a K-medoids clustering algorithm to select datapoints for active learning. Their method greatly outperforms the baselines.

**Audience:**

Yes

**Claims And Evidence:**

Yes

**Requested Changes:**

- I don't think the authors perform repeated runs of the method on the algorithm. The authors use 4 digits of standard deviation, which does not mean much without repeated runs. Two digits significance should be sufficient.
- Section 3.1: I found the part below equation 1 hard to understand, but it is quite vital. May be worth spending more words on the different sets introduced here, and what is meant with 'selecting' subsets of nodes and what the annotators do. (Eg what is the goal of this selection procedure).
- Section 3.3: My understanding is that the authors implemented the OOD filter by training a C+1-label classifier (where the last class corresponds to OOD). Why not use a binary classifier for in vs out of distribution?
- Section 3.6: I was confused that the ID classifier is used for OOD detection post-hoc. Why not the OOD filter?
- Section 4.1.1: Is the split between ID and OOD classes in some way meaningful? Or just random? It would be nice for the story of the paper if there's a setup like Figure 1 / page 2 where there is a clear distinction.
- Figure 3 is pretty confusing to me design-wise. For the same space, a table should just work? And the use of averaging of the other methods is also confusing to me.
- Section 4.3: Table 8 is referenced but I think it should be table 7.

**Strengths And Weaknesses:**

**Strengths**: This paper seems solid to me. I was able to understand the main points and ideas fairly quickly. The experimental evaluation seems fairly thorough with new baselines which it outperforms on 4 datasets. There is a nice ablation study.

**Weaknesses**:
- At points, the writing can be a bit sharper, clearer and to the point.
- The experiments lack repeated runs
- The experimental setup is not as clearly compelling as the given example

---

> ### Author Response · Authors · 2025-03-23
>
> **Weakness 2 & request 1**: Thank you for pointing this out. The four-digit values are variance results, so we have updated them to standard deviation values with two digits for the 10 repeated experiments, as shown in the table below. In this paper, all experimental results are obtained from 10 repeated runs, averaging across 10 different random seeds.
>
> | STD | ID ACC | AUROC  | AUPR | FPR80 |
> | -------- | -------- | -------- | -------- | -------- |
> | Cora (LEGO-Learn)| 6.54% | 1.40% | 0.30% | 13.62%|
> | LastFMAsia (LEGO-Learn)     | 3.40% | 3.13% | 2.44% | 10.23%|
> | Cora (GCN-ENT)      | 6.30%  |  5.46%  |  5.63%  |  14.92% |
> | LastFMAsia (GCN-ENT)      | 4.25% | 2.89% | 3.31% | 12.62%|
>
>
> **Weakness 1 & Request 2**: Thank you for your valuable feedback. We appreciate your observation that the explanation below Equation (1) could be clearer, especially since it plays a crucial role in understanding our method. In the revised version, we have added clarifying details in Section 3.1 to explicitly define the different node sets involved—namely, the labeled set $\mathcal{V}_L$, unknown (OOD) set $\mathcal{V}_O$, and the unlabeled pool $\mathcal{V}_U$.
> Specifically, let $\mathcal{V}_L$ be the current labeled set, $\mathcal{V}_U$ the pool of unlabeled nodes available for selection, and $\mathcal{V}_O$ the set of nodes identified as OOD, i.e., not belonging to any known ID class.
>
>
> We also clarify the goal of the selection procedure, which is to select a subset of nodes $\mathcal{V}^s \subset \mathcal{V} \setminus \mathcal{V}_{\text{train}}$ under a label budget constraint, such that querying their labels will allow us to construct a strong ID classifier.
> For the annotators, they provide ground-truth class labels only for the selected nodes. This selective annotation is essential for maintaining label efficiency within the given budget.
>
> **Weakness 3**: Thank you for your insightful comment. Real-world datasets are not as well-divided as shown in Fig. 1 in the paper, but LEGO-Learn still performs well on the four datasets which have more complex graph structure as shown in the experiments.
>
> **Request 3**: Thank you for your constructive feedback. We can also use a binary classifier as the OOD filter to differentiate between ID and OOD nodes before human annotation. This time, we simply use a C+1 classifier as the OOD filter and show its effectiveness. We will explore more effective filters in future studies.
>
>
> **Request 4**: Thank you for sharing your thoughts. In this paper, our main goal is to train a robust, label-efficient ID classifier under an open-set, low-resource setting. As a result, although the OOD filter can also perform OOD detection, its performance is not our primary concern. The purpose of the OOD filter is to remove potential OOD nodes before human annotation, thereby reducing labeling costs. After training the ID classifier, we can apply post-hoc OOD detection methods on top of it to enhance its OOD awareness in real-world open-set scenarios. In summary, although the OOD filter has the ability to detect OOD nodes, our ultimate goal and main takeaway is to train a robust ID classifier that performs well in both ID classification and OOD detection.
>
>
> **Request 5**: Thank you for the insightful question. We follow the same OOD category division setting as described in [a]. To demonstrate generalizability, we also modify the split between ID and OOD classes on Cora (with ID classes now being 1, 2, 3, and 4) and LastFMAsia (with ID classes now being 0 to 8). The results are presented in Appendix E.
>
> [a] Song, Yu, and Donglin Wang. "Learning on graphs with out-of-distribution nodes." KDD 2022.
>
>
> **Request 6**: We sincerely appreciate your insightful feedback. In the figure, for each node selection method, we averaged the precision results across different graph open-set learning models. This is because current graph open-set learning methods do not explicitly consider label efficiency and thus produce similar precision values under the same node selection strategy. In contrast, different node selection methods can yield significantly different precision results. Thank you again for your comments—we agree that a table is more space-efficient. In the revised version, we have replaced the figure with a table.
>
>
> **Request 7**: We have corrected the table reference.

---

### Review · Reviewer_RfXH · 2025-03-12

**Summary Of Contributions:**

The paper introduces LEGO-Learn, a framework for label-efficient graph open-set learning (GOL). The proposed method addresses the challenge of training graph-based models to recognize unseen classes while minimizing labeling costs. LEGO-Learn employs a GNN-based filter to identify and exclude potential out-of-distribution (OOD) nodes and selects highly informative in-distribution (ID) nodes for labeling using the K-Medoids algorithm. The framework is evaluated on four real-world datasets, demonstrating significant improvements in ID classification accuracy and OOD detection compared to existing methods.

**Audience:**

Yes

**Claims And Evidence:**

Yes

**Requested Changes:**

- What is the real-world significance of this new setting?
- Why can GNNs filter out most of the OOD nodes? Is there any theoretical guarantee for this? As is well known, GNNs tend to suffer from over-smoothing. I have concerns about their ability to identify outliers.
- As is well known, a significant portion of node anomalies stems from the influence of their topological structure. Apart from using GNNs, the paper does not incorporate any additional graph structural information. Can this approach accurately identify anomalous nodes?
- Is there any visualization to demonstrate that the authors have indeed identified anomalous nodes and accurately predicted in-distribution (ID) nodes?
- Can the authors analyze the effectiveness of their method from multiple perspectives, such as complexity analysis, efficiency comparison, sensitivity analysis, and case studies, to further enrich their experiments?

**Strengths And Weaknesses:**

- The paper tackles a problem in graph-based learning, offering a novel solution that integrates OOD filtering and label-efficient node selection.
- Extensive experiments on four datasets show that LEGO-Learn outperforms existing methods in both ID classification and OOD detection.

---

> ### Author Response · Authors · 2025-03-23
>
> **Request 1**: Thank you for your insightful question. Our method focuses on label-efficient graph open-set learning, a crucial problem in real-world applications where labeled data is limited and unseen classes frequently appear during inference.
> A practical example is: Consider a social network where nodes represent individuals, node attributes correspond to their textual descriptions, and edges denote interactions or connections between them. Initially, the entire network is unlabeled, and the goal is to classify individuals into specific interest groups, such as technology enthusiasts, sports fans, or musicians, while operating within a limited human annotation budget. However, the network also contains individuals whose interests fall outside these predefined categories, such as those primarily engaged in political discussions or travel blogging. Identifying and labeling these OOD nodes would be inefficient, as they do not contribute to training an effective classifier for the targeted interest groups. Instead, the focus is on accurately classifying only the ID nodes while detecting and filtering out OOD nodes that do not belong to the intended classification space. With our proposed method, we can train the ID classifier more efficiently under a label budget constraint. In other words, our method is more effective in open-set, low-resource settings.
>
>
> **Request 2**: Thank you for the in-depth comment on the effectiveness of the GNN filter. Our GNN filter can remove some OOD nodes, although it may not always filter out most of them. However, as shown in Section 4.3, after applying the GNN filter, the proportion of ID nodes has indeed improved significantly. This is because we extend it to the (C+1)-th classifier, and the nodes predicted as the (C+1)-th class are more likely to be OOD nodes. Nevertheless, we acknowledge concerns regarding over-smoothing in deep GNNs. To mitigate this, we use only shallow GNNs (2 layers) to prevent over-smoothing.
> While there is no formal theoretical guarantee, our experimental results demonstrate that GNNs can serve as effective OOD filters. Future work could explore designing more advanced OOD filters and potentially leveraging LLMs for zero-shot OOD detection and filtering of OOD nodes.
>
>
> **Request 3**: Thanks for sharing your thoughts. We agree that a node's topological structure is critical for identifying anomalies. Our framework consists of a GNN filter, an ID classifier, a node selection module, and a post-hoc OOD detector. The filter and ID classifier are based on GNNs, while the node selection module leverages a graph active learning method. Additionally, we can utilize graph-specific post-hoc OOD detectors for OOD detection. As you can see, all four modules in our framework are specifically designed for graph data.
>
>
> **Request 4**: We visualize the OOD scores predicted by LEGO-Learn for ID and OOD nodes in the test set in Appendix G. As shown in the figure, the scores for ID and OOD inputs are well-separated. This proves that our method effectively identifies OOD nodes and accurately predicts ID nodes.
>
>
> **Request 5**: We appreciate this suggestion and provide additional analysis, including:
> Complexity Analysis: The GNN filter and ID classifier have a complexity of $O(E)$, where E is the number of edges. Node selection with clustering has a complexity of $O(N\times k)^2$, where $k$ is the ratio of the number of potential ID nodes to the total number of nodes in the graph.
> Sensitivity Analysis: We analyze the impact of filter weight $w$ in the weighted cross-entropy loss in our GNN filter in equation 4 and appendix F in the paper.
> We present the ID classification accuracy of our method under different w values in the table below for LastFMAsia.
>
> |w|0.01|0.1|0.2|0.3|0.4|1|
> |-|-|-|-|-|-|-|
> |LastFMAsia|0.7624|0.7800|0.7770|0.7692|0.7650|0.7602|

---

### Review · Reviewer_598D · 2025-03-16

**Summary Of Contributions:**

This paper is (very likely) the first attempt to address the important but under-explored problem of label-efficient graph open set learning (GOL). The authors propose LEGO-Learn, a two-stage framework that combines a GNN-based filter for OOD removal with a K-Medoids clustering strategy for selecting diverse and informative ID nodes under a strict label budget.

**Audience:**

Yes

**Claims And Evidence:**

Yes

**Requested Changes:**

1. The first recommended change is adding openness ratio experiments. I strongly recommend the authors conduct experiments that vary the ratio between OOD and ID classes in the test set (i.e., degree of “open-world-ness”). This will help assess how LEGO-Learn performs across different levels of openness, from low-OOD to highly OOD-dominated scenarios. Without this, it is hard to fully validate the framework’s generalizability to varying real-world open-set conditions.

2. To fully contextualize LEGO-Learn’s performance, please include experiments at the two boundary cases: (a) Fully closed-world (all test nodes are ID) to compare against classical node classification methods. (b) Fully open-world (all test nodes are OOD) to compare against classical OOD detection methods.

Then, the progressive levels between (a) and (b) would also help position LEGO-Learn relative to both standard GNN classifiers and graph OOD baselines.

3. Since the paper positions LEGO-Learn as a label-efficient solution, I recommend providing experiments that vary the amount of available annotated data (e.g., from 1% to 10% labeled nodes). This will better showcase the method’s performance under different levels of supervision.

4. While optional, it would significantly strengthen the paper to visualize or discuss the trade-offs LEGO-Learn faces as the openness ratio or label budget changes. This could help practitioners understand when and how to deploy LEGO-Learn based on resource availability and the expected level of open-worldness in their graphs.

**Strengths And Weaknesses:**

Strengths:
1. The paper tackles a practically important and underexplored problem—label-efficient graph open-set node classification—which is highly relevant to real-world scenarios where annotation costs are prohibitive, and OOD classes commonly exist.

2. The implementation is natural for its target: by combining a GNN-based OOD filter with a K-Medoids-based node selection strategy, the paper proposes a complete and principled pipeline that handles both selective annotation and open-set filtering, ensuring that both classification accuracy and OOD detection performance are addressed simultaneously.

Weaknesses:
1. The paper does not provide experiments that systematically vary the ratio between unseen (OOD) classes and known (ID) classes to measure how the method performs under different levels of “openness.” In practice, the degree of open-world-ness can significantly impact both ID classification and OOD detection performance. Without such experiments, it is difficult to assess how well LEGO-Learn manages the trade-off between filtering OOD nodes and retaining informative ID nodes as this ratio shifts.

2. The paper could benefit from evaluating the method in fully closed-world (all test nodes are ID) and fully open-world (all test nodes are OOD) settings. These extremes could provide meaningful anchors for comparison against classical node classification methods (in closed-world) and classical OOD detection methods (in fully open-world). This would help clarify LEGO-Learn’s relative performance in scenarios where specialized baselines may excel.

3. Given that the paper emphasizes label-efficiency, it is surprising that no experiments systematically vary the available annotation ratio. Varying the amount of labeled data (e.g., from very scarce labels to moderate label availability) would help demonstrate LEGO-Learn’s robustness under different supervision levels and better support its label-efficient claims.

4. The framework likely involves a trade-off between ID classification accuracy and OOD detection performance as openness increases, and potentially another trade-off between model performance and label availability. However, these trade-offs are not discussed or visualized in the paper (correct me if I missed). A systematic exploration of these dynamics could make the contributions more convincing and generalizable.

---

> ### Author Response · Authors · 2025-03-23
>
> **Weakness 1 & Request 1**: Thank you for this insightful suggestion. To analyze how LEGO-Learn performs under different degrees of openness, we conduct experiments on the Cora dataset by varying the OOD ratio in the test set, as shown below. Specifically, we test the model with OOD ratios ranging from 10% to 90% in increments of 10%. We run all experiments 10 times and average the results across 10 random seeds. The results show that LEGO-Learn consistently outperforms the baseline across all openness levels.
>
> |ID nodes proportion|**Baseline** AUROC↑|AUPR↑|FPR95↓|**LEGO-Learn** AUROC↑|AUPR↑|FPR95↓|
> |-|-|-|-|-|-|-|
> |10%|0.7950|0.3861|0.6601|0.8355|0.5386|0.4682
> |20%|0.7870|0.5284|0.6685|0.8418|0.6749|0.4598
> |30%|0.7812|0.6308|0.6764|0.8306|0.7444|0.4809
> |40%|0.7893|0.7218|0.6716|0.8222|0.7980|0.4876
> |50%|0.7876|0.7919|0.6543|0.8380|0.8600|0.4665
> |60%|0.7833|0.8412|0.6773|0.8335|0.8906|0.4900
> |70%|0.7866|0.8904|0.6587|0.8298|0.9210|0.4908
> |80%|0.7832|0.9291|0.6691|0.8300|0.9504|0.5106
> |90%|0.7889|0.9668|0.0000|0.8247|0.9753|0.0000
>
>
> **Weakness 2 & Request 2**: Thank you for this great suggestion. In the fully closed-world setting, 100% of the nodes in the test set are ID nodes. In this case, we can only perform ID classification, not OOD detection, using all the ID nodes in the test set. As a result, the ID classification performance of our method and the baseline is shown in the following table. As we can see from the results, our method outperforms the baseline because it selects more potential ID nodes for labeling and training the ID classifier.
>
> |Model|Cora|Amazon-CS|Amazon-photo|LastFMAsia|
> |-|-|-|-|-|
> |GCN-ENT|0.8254|0.7420|0.9054|0.6902|
> |LEGO-Learn|0.8684|0.8710|0.9648|0.7818|
>
> In the fully open-world setting, if 100% of the nodes in the test set are OOD nodes, the common metrics (AUROC and AUPR) for graph OOD detection become invalid. This is because both AUROC (Area Under the Receiver Operating Characteristic Curve) and AUPR (Area Under the Precision-Recall Curve) require at least two different labels (i.e., both positive and negative classes) to be valid. Since graph OOD detection typically work in the transductive setting, where the entire set of nodes is accessible during training, but only a portion of the class labels (ID classes) is provided. In this paper, we follow the common setting [a] in graph OOD detection and use metrics such as AUROC and AUPR to evaluate the performance of different methods. A potential future direction is to extend the ID classifier for joint classification, where performance can be evaluated using weighted-F1. In this task, we address cases where 100% of the nodes in the test set are OOD nodes by treating ID classification and OOD detection as a multi-class classification problem with N+1 classes, i.e., N in-distribution classes and one OOD class.
>
> [a] Wu, Qitian, et al. "Energy-based out-of-distribution detection for graph neural networks." ICLR 2023.
>
>
> **Weakness 3 & Request 3**: Thank you for sharing your insightful opinion. We vary the amount of available annotated data by changing the label budget to 10×C, 15×C, and 20×C, where C is the number of ID classes. The corresponding ratios of labeled nodes are 1.18%, 1.77%, and 2.36%, respectively. Due to space limitations, we present the results for different label budgets on the LastFMAsia dataset in Appendix C.
>
>
> **Weakness 4 & Request 4**: For all methods, including baselines and LEGO-Learn, increasing the openness ratio in the graph generally requires a higher label budget to achieve similar performance when training the ID classifier. Additionally, increasing the label budget improves both ID classification and OOD detection performance. In this paper, we focus on the data-efficient case, where the given label budget is small. However, our method does not face significant trade-off challenges across different openness ratio scenarios. This is because our designed OOD filter effectively removes many potential OOD nodes before node selection for human annotation. Even if the graph initially contains many OOD nodes, we can still select a high proportion of ID nodes to train our target classifier. In contrast, all current baseline methods are significantly affected by different levels of openness.

---

### Decision · Action_Editor_NGss · 2025-04-16

**Recommendation:** Accept as is

**Comment:**

All reviewers agree that the paper should be accepted. I think that the revisions cleared the concerns raised by reviewers, and that the paper is interesting and valuable to the community.

**Audience:**

All reviewers agree that this submission is relevant to the audience of TMLR and I support this view as well.

**Claims And Evidence:**

All reviewers agree that the claims are supported by the evidence. I agree with that.